# Antimicrobial Stewardship Improvement in Pediatric Intensive Care Units in Spain—What Have We Learned?

**DOI:** 10.3390/children9060902

**Published:** 2022-06-16

**Authors:** Elena Fresán-Ruiz, Ana Carolina Izurieta-Pacheco, Mònica Girona-Alarcón, Juan Carlos de Carlos-Vicente, Amaya Bustinza-Arriortua, María Slocker-Barrio, Sylvia Belda-Hofheinz, Montserrat Nieto-Moro, Sonia María Uriona-Tuma, Laia Pinós-Tella, Elvira Morteruel-Arizcuren, Cristina Schuffelmann, Yolanda Peña-López, Sara Bobillo-Pérez, Iolanda Jordan

**Affiliations:** 1Pediatric Intensive Care Unit, Hospital Sant Joan de Déu, University of Barcelona, 08950 Barcelona, Spain; efresan@hsjdbcn.es (E.F.-R.); mgirona@hsjdbcn.es (M.G.-A.); sbobillo@hsjdbcn.es (S.B.-P.); 2Immunological and Respiratory Disorders in the Pediatric Critical Patient Research Group, Institut de Recerca Sant Joan de Déu, University of Barcelona, 08950 Barcelona, Spain; 3Pediatrics Department, Hospital Sant Joan de Déu, University of Barcelona, 08950 Barcelona, Spain; acizurieta@hsjdbcn.es; 4Pediatric Intensive Care Unit, Hospital Son Espases, 07120 Palma de Mallorca, Spain; juanc.decarlos@ssib.es; 5Pediatric Intensive Care Unit, Hospital Gregorio Marañón, 28007 Madrid, Spain; amayabustinza@outlook.es (A.B.-A.); mslocker@gmail.com (M.S.-B.); 6Pediatric Intensive Care Unit, Hospital 12 de Octubre, 28041 Madrid, Spain; sylvia.belda@salud.madrid.org; 7Pediatric Intensive Care Unit, Hospital Niño Jesús, 28009 Madrid, Spain; mnietom@salud.madrid.org; 8Preventive Medicine and Public Health, ENVIN-HELICS Registry Administration, Hospital Vall d’Hebron, 08035 Barcelona, Spain; smuriona@vhebron.net (S.M.U.-T.); lpinos@vhebron.net (L.P.-T.); 9Pediatric Intensive Care Unit, Hospital de Cruces, 48903 Bilbao, Spain; elvira.morteruelarizcuren@osakidetza.net; 10Pediatric Intensive Care Unit, Hospital La Paz, 28046 Madrid, Spain; cschuffelmann@yahoo.es; 11Pediatric Intensive Care Unit, Hospital Materno-Infantil Vall d’Hebron, 08035 Barcelona, Spain; ypena@vhebron.net; 12Consorcio de Investigación Biomédica en Red de Epidemiología y Salud Pública (CIBERESP), 28029 Madrid, Spain

**Keywords:** antimicrobial stewardship, antibiotics, resistant microorganisms, de-escalation, early suspension, pediatric intensive care, children

## Abstract

Antibiotic misuse in pediatric intensive care units (PICUs) can lead to increased antimicrobial resistance, antibiotic-triggered side effects, hospital costs, and mortality. We performed a multicenter, prospective study, analyzing critically ill pediatric patients (≥1 month to ≤18 years) admitted to 26 Spanish PICUs over a 3-month period each year (1 April–30 June) from 2014–2019. To make comparisons and evaluate the influence of AMS programs on antibiotic use in PICUs, the analysis was divided into two periods: 2014–2016 and 2017–2019 (once 84% of the units had incorporated an AMS program). A total of 11,260 pediatric patients were included. Total antibiotic prescriptions numbered 15,448 and, overall, 8354 patients (74.2%) received at least one antibiotic. Comparing the two periods, an increase was detected in the number of days without antibiotics in patients who received them divided by the number of days in PICUs, for community-acquired infections (*p* < 0.001) and healthcare-associated infections (HAIs) acquired in PICUs (*p* < 0.001). Antibiotics were empirical in 7720 infections (85.6%), with an increase in appropriate antibiotic indications during the second period (*p* < 0.001). The main indication for antibiotic adjustment was de-escalation, increasing in the second period (*p* = 0.045). Despite the high rate of antibiotic use in PICUs, our results showed a significant increase in appropriate antibiotic use and adjustment following the implementation of AMS programs.

## 1. Introduction

Undoubtedly, antibiotic resistance has become one of the greatest threats of our time. Not 100 years have passed since the discovery of antibiotics transformed healthcare practices, heralding an abrupt deceleration of mortality coinciding with the first clinical use of sulfonamides in 1935 and penicillin in 1942 [1]. Nevertheless, we already feel threatened by the rise of multi-resistant bacteria as a consequence of long-standing antibiotic misuse and overuse [2]. Not long ago, this emerging problem seemed to have been solved with the development of new antimicrobials, but these options, too, are quickly becoming less effective. In 2008, over 25,000 people in Europe alone died as a result of serious infections caused by multi-drug-resistant bacteria, and if this problem is not effectively remedied, the situation is expected to worsen over the coming decades [3]. The problem is especially evident in intensive care units, where, in dealing with critically ill patients, there is an extensive use of broad-spectrum antibiotics during prolonged periods, use of invasive devices, and, consequently, increases in the numbers of multi-drug-resistant bacteria causing severe infections [4].

In recent years, antimicrobial stewardship (AMS) programs have become the cornerstone of slowing the development of antimicrobial resistance, optimizing not only the treatment of infections but also reducing adverse effects that result from the overuse of antimicrobials, as well as reducing the associated costs [5,6]. Several strategies have been implemented in PICUs in Spain since 2014 in order to ensure effective AMS programs, including preauthorization, prospective audits, feedback interventions on antimicrobial prescriptions, the development of empirical treatment and prophylaxis guidelines, as well as physician education and consolidation of AMS teams with a continuous monitoring of antimicrobial use [3,7].

In today’s challenging environment, with no immediate prospects for any new broad-spectrum antibiotics, especially against enterobacteria, and an aging and/or compromised patient population, the implementation of AMS practices has shown to decrease antimicrobial use by 22–36%, as observed in both large and small community hospitals [8]. In Spain, a nationwide PROA was launched in 2012 as part of the National Antibiotic Resistance Plan [9] in an effort to preserve the effectiveness of antimicrobial agents and in response to the increasing bacterial resistance in hospital environments associated with poorer clinical outcomes as well as elevated healthcare costs. 

Furthermore, with a view to developing a surveillance system for healthcare-associated infections (HAI), a multicenter registry of HAIs in Spanish Pediatric Intensive Care Units (PICUs) was created in 2007 [10]. This was consolidated in 2013 as the National Nosocomial Infections Surveillance System (ENVIN, in Spanish) within the HELICS (Hospitals in Europe Link for Infection Control through Surveillance) project [11,12]. 

The data included in this far-reaching registry provide important information not only about the overall rates of HAI but also about the use of antibiotics, the microorganisms isolated, and the resistance profile. It is considered a national and international benchmark for HAIs in PICUs. This study offers novel information for use in the field, data on the use of broad-spectrum antibiotics and HAIs being scarce in the literature [13,14], as are data regarding the influence of AMS programs in national PICUs. In this study, we aimed to describe the use of antibiotics in the PICUs participating in the Pediatrics-ENVIN-HELICS multicenter registry and then compared their evolution over time, considering the hypothesis that the implementation of AMS programs may have improved antibiotic use. 

## 2. Materials and Methods

A prospective, multicenter observational study was conducted in 26 PICUs in Spain, over a 3-month period each year (from 1 April to 30 June, as per ENVIN-HELICS surveillance criteria) from 2014 to 2019. The reason for choosing these particular months was that it is a time of the year with a medium workload, so it is estimated that the data are more accurate.

The subjects included were hospitalized pediatric patients (≥1 month and ≤18 years of age) who required admission to the PICU during the study period. All patients were admitted before the study period began (1 April) and those who were hospitalized in the PICUs after 30 June were excluded.

Patients registered by all of the PICUs participating in the National Nosocomial Infections Surveillance System were included in this study. However, at the beginning of the data collection period, not all of the PICUs had implemented an AMS program. We conducted a recurring survey among these 31 units in order to determine exactly when they did implement an AMS program. By the end of the study period, 16% (*n* = 5) of them had not yet done so. By 2016, only 55% (*n* = 17) of the units had established one of these programs. However, by 2019, 84% (*n* = 26) of the PICUs had incorporated an AMS program. To test the hypothesis that AMS programs may influence antibiotic use, the analysis was divided into two 3-year periods, 2014–2016 and 2017–2019, for comparison.

### 2.1. Definitions

Comorbidities: These were considered as the presence of any of the following underlying diseases: diabetes, kidney failure, immunosuppression, neoplasia, cirrhosis, chronic obstructive pulmonary disease, malnutrition, and transplantation. 

Healthcare-associated infections [15,16]: HAIs were differentiated between those contracted outside and inside the PICU, depending on where the infection started. We express the rate of HAI as HAIs/1000 patient-days. Patient-days, as their name suggests, are defined as the number of days that patients spent in PICUs. 

A device-related HAI was considered in a patient with a device (endotracheal tube, central line, or indwelling urinary catheter) that was used within the 48 h period before the onset of the infection, even if it was only used intermittently [17,18]. Table 1 includes the different definitions of infections: ventilator-associated pneumonia (VAP), central line-associated bloodstream infection (CLABSI), and catheter-associated urinary tract infection (CAUTI).

### 2.2. Outcome

The primary outcome was to determine changes in the use of antibiotics over time, considering the influence of the implementation of AMS programs. The antibiotic use indicators considered were:

Outside-PICU antibiotic use: Antibiotics prescribed for infections diagnosed once the patient was hospitalized for more than 48 h in a hospital ward.

Inside-PICU antibiotic use: Antibiotics prescribed for infections diagnosed once the patient was hospitalized for more than 48 h in the PICU.

Antibiotic-free days: The difference between the duration of the antibiotic course(s) and the length of stay in the PICU. 

Antibiotic use ratio: The number of patients who received antibiotics with respect to the total number of patients included in the study. 

Antibiotic-free days ratio: The number of days without antibiotics in patients who received them divided by the number of days these patients spent in the PICU. 

Global antibiotic-free days ratio: The number of days without antibiotics, considering all the patients with respect to the overall number of days they were in the PICU [11].

### 2.3. Variables

Physicians at the participating hospitals entered all the clinical data for each patient admitted to the PICU during the study period into a standardized online registry. The parameters collected for this analysis included demographic characteristics, pediatric risk of mortality (PRISM) score and Glasgow score at admission, comorbid medical conditions, predisposing factors for infections, diagnosis at admission, community-acquired infections, HAIs acquired outside and inside the PICU, device-associated HAIs, microbiological data, length of stay (LOS), antibiotics used, and antibiotic susceptibility profile [10,11].

### 2.4. Statistical Analysis

The statistical analysis was performed using SPSS 25.0 Statistics^®^, IBM (1 New Orchard Road Armonk, New York 10504-1722 United States). Categorical variables were indicated as frequency (n) and percentage (%), while continuous variables were summarized as median and interquartile range (IQR) because they were not normally distributed. The comparison of qualitative variables was performed using the χ^2^ test or Fisher’s exact test. Continuous variables were compared using the Mann–Whitney U test when not normally distributed. Probability values of less than 0.05 were considered statistically significant.

## 3. Results

### 3.1. Patients and Clinical Characteristics

A total of 11,260 pediatric patients were included in the registry from 2014 to 2019. Among them, 56.6% (*n* = 6368) were males and the median age was 43.2 months (*IQR* 10–115 months). Table 2 includes a general description of the sample by year and also data regarding the evolution. Of the patients included, 2.8% (*n* = 317) presented at least one device-related HAI during their stay in the PICU. The total number of HAIs was 390, and the overall rate of HAI was 6.3 per 1000 patient-days. The most frequent device-related HAI was VAP, with 110 episodes, followed by CAUTI, with 105 episodes, and CLABSI, with 90 episodes.

### 3.2. General Use of Antibiotics

The total number of antibiotic prescriptions was 15,448 and, overall, 8354 patients (74.2%) received at least one antibiotic during their PICU stay. The ratio of antibiotics per patient prescribed antibiotics, meaning the average number of antibiotic prescriptions given to each patient who required them, showed a progressive decrease until 2019, when it skyrocketed to 2014 levels (Figure 1).

Surgical prophylaxis was the main indication for antibiotics, with a total of 5024 (32.5%) antibiotics used. Community-acquired infection was the second most common antibiotic indication, with 4690 (30.4%) antibiotic prescriptions. The suspicion of a HAI acquired inside the PICU was the third most common reason for prescribing antibiotics (*n* = 2201, 14.2%), followed by the suspicion of a HAI acquired outside the PICU (*n* = 2122, 13.7%), other non-surgical prophylaxis (*n* = 1299, 8.4%), and unknown reason for prescription (*n* = 112, 0.70%). Upon comparing these antibiotic indications between the time periods (2014–2016 vs. 2017–2019), the only statistically significant differences detected were among HAIs acquired outside the PICU (14.1% vs. 13.3%, *p* < 0.001) and unknown reasons for prescription (0.99% vs. 0.47%, *p* < 0.001). The detailed results regarding the number of antibiotics prescribed for each indication are included in Table 3.

Figure 2 represents the different ratios regarding the general use of antibiotics: antibiotic use ratio, antibiotic-free days ratio, and global antibiotic-free days ratio.

Table 4 summarizes the details by type of suspected infection, with a comparison between the two data groups. As shown, in the second period, a statistically significant increase was detected for the number of antibiotics per patient prescribed antibiotics for inside-PICU-acquired HAIs. In addition, statistically significant differences were globally detected in terms of the antibiotic use ratio between the two groups, although these differences were not confirmed for community-acquired infections, outside-PICU-acquired HAIs, or inside-PICU-acquired HAIs. Moreover, a statistically significant increase was detected for the antibiotic-free days ratio and the global antibiotic-free days ratio in the second period for community-acquired infections and inside-PICU-acquired HAIs.

### 3.3. Types of Antibiotics Prescribed

Table 5 summarizes the antibiotics most commonly used for the different indications. An increase was observed in the use of cefazoline for surgical prophylaxis between the two periods, this antibiotic being the most frequently used for this indication. The most widely used antibiotic in community-acquired infections was cefotaxime, followed by amoxicillin–clavulanate, with no differences between the two periods of time compared. In the case of suspected outside-PICU-acquired HAIs, a statistically significant increase in the use of vancomycin and teicoplanin was observed, with a parallel decrease in the use of piperacillin–tazobactam. The empirical prescription of antibiotics for inside-PICU-acquired HAIs was similar between the two periods, except for the increase in the prescription of teicoplanin in the 2017–2019 period.

Meropenem represented 5.4% of the total antibiotics prescribed. The global use of meropenem remained stable over time, but the indication of meropenem for suspected inside-PICU-acquired HAIs varied, as shown in Figure 3. The indication of meropenem for outside-PICU-acquired HAIs remained stable until 2018, when an increase was observed, and it subsequently decreased, reaching its lowest value in 2019.

### 3.4. Empirical Use of Antibiotics in Suspected Infections

There was a total of 9013 courses of antibiotics prescribed due to suspected infections. The indication of antibiotic therapy was empirical in 7720 suspected infections (85.6%) and in 1219 (13.5%) cases was pathogen-targeting therapy, with no differences between the two periods (*p* = 0.147 and *p* = 0.230, respectively). Regarding suspected community-acquired infections, 4310 antibiotics (92.7%) were empirical and 340 (7.3%) were pathogen-targeting. As for suspected outside-PICU-acquired HAIs, 1726 (82.1%) of the prescriptions were empirical and 375 (17.8%) were pathogen-targeting. Finally, as regards suspected inside-PICU-acquired HAIs, 1684 (77%) of these antibiotics were empirical and 504 (23%) were pathogen-targeting therapies. There were statistically significant differences between the three groups and the empirical or pathogen-targeting therapies, with all *p*-values < 0.001. 

Empirical therapy was considered appropriate when microorganisms were susceptible in vitro to antibiotics. In this sample, the empirical prescriptions were appropriate in 24.9% of cases (*n* = 1972) and the distribution was similar in community-acquired infections (24.7%), outside-PICU-acquired HAIs (23.4%), and inside-PICU-acquired HAIs (26.7%). Empirical antibiotic therapy was considered inappropriate in 12.1% of all cases (*n* = 957): 11.3%, 12.3%, and 13.8%, respectively, for the aforementioned groups. No statistically significant differences were observed as regards the appropriate or inappropriate prescription and the type of suspected infection (community-acquired, outside-PICU-acquired, and inside-PICU-acquired HAIs), *p* = 0.231. In 45.7% of the indications (*n* = 3621), negative cultures were found: 45.5% in community-acquired infections, 47.9% in outside-PICU-acquired HAIs, and 43.8% in inside-PICU-acquired HAIs, with statistically significant differences between them (*p* = 0.011).

An increase in appropriate antibiotic indications was detected when comparing the two periods, as shown in Figure 4 (Plot 1), with statistically significant differences (*p* < 0.001) and a parallel significant decrease in negative cultures (*p* < 0.001). No statistically significant differences were detected for inappropriate indications (*p* = 0.162).

In up to 65.7% of the antibiotics indicated for empirical treatment (*n* = 5512), no modifications were made and 20.2% were suspended early, while an antibiotic adjustment was made in 14.2% of the prescriptions. No statistically significant changes were detected over time regarding antibiotic adjustment, early suspension, or absence/presence of a change in antibiotics. Figure 3 (Plot 2) details the evolution of these parameters over time. Antibiotic adjustments were more frequent for inside-PICU-acquired HAIs (20.4%) than for outside-PICU-acquired HAIs (15.5%) and community-acquired infections (10.7%), with *p* < 0.001. Early suspension was more frequent in suspected community-acquired infections (21.8%) than in inside-PICU-acquired HAIs (20.5%) and outside-PICU-acquired HAIs (16.1%), with *p* < 0.001.

The main indication of the 1,189 antibiotic adjustments was reducing the antibiotic spectrum in favor of a narrower one (*n* = 485, 40.8%), followed by a poor clinical evolution (*n* = 349, 29.4%) and a need to increase the antibiotic spectrum (*n* = 159, 13.4%). Figure 5 represents the evolution over time regarding the different reasons for switching antibiotics. Upon comparing the two periods, statistically significant differences were detected regarding the reduction of the antibiotic spectrum (2014–2016, 37.6% vs. 2017–2019, 43.3%; *p* = 0.045), without clear differences regarding the other items: increasing the spectrum, 13.7% vs. 12.9%, *p* = 0.696; poor clinical outcome, 31.9% vs. 27%, *p* = 0.062; new resistances, 1.6% vs. 0.8%, *p* = 0.227; and toxicity, 3.5% vs. 4.4%, *p* = 0.426.

Reducing the antibiotic spectrum was more frequent for inside-PICU-acquired HAIs (44%) than for community-acquired infections (39.9%) and outside-PICU-acquired HAIs (37.7%), with *p* = 0.049. 

Out of the 9013 antibiotic courses prescribed for a suspected infection, de-escalation was practiced in 5.4% (*n* = 485) of the cases. Antibiotic adjustment due to poor clinical evolution was more frequent in community-acquired infections (32.9%) and outside-PICU-acquired HAIs (32.8%) than in inside-PICU-acquired HAIs (22.8%), with *p* = 0.004. The increase of the antibiotic spectrum was more frequent in inside-PICU-acquired HAIs (17.5%) and outside-PICU-acquired HAIs (14.9%) than in community-acquired infections (8.7%), with *p* = 0.001.

## 4. Discussion

To the best of our knowledge, this is the first prospective multicenter study conducted in PICUs that provides an in-depth evaluation of the total antibiotic consumption for both community-acquired infections as well as HAIs. Our results demonstrate a significant increase in appropriate antibiotic use and also a reduction in antibiotic utilization, either by early suspension or antibiotic treatments being adjusted to target a narrower spectrum, attributable to the implementation of AMS programs. 

Despite efforts to control antimicrobial use, the number of antibiotic prescriptions written for hospitalized children continue to increase. In 2008, Gerber et al. [19] reported, using a retrospective cohort from 40 independent children’s hospitals in the US, that 60% of children received at least one antibiotic during their hospitalization, with an average of 468/1,000 inpatient-days spent on antibiotics. Similarly, in 2016, the Global Point Prevalence Survey showed that at least 37% of children were given an antibiotic in pediatric inpatient settings, with the highest antimicrobial use being up to 61% in the PICU [20,21,22]. This was also reflected in the European Surveillance of Antimicrobial Consumption point prevalence survey in 2008, which demonstrated that at least 32% of children received one or more antimicrobials, with third-generation cephalosporins as the leading therapeutical group in pediatric units [23].

Notably, in our sample, up to 74.2% of patients admitted to PICUs received at least one antibiotic. The leading indication was surgical prophylaxis; therefore, an effort needs to be made to decrease the duration of this antibiotic prescription, according to international guidelines. Nevertheless, it is likely that the higher proportion of antibiotic use that we detected may be influenced by the high complexity of the care centers included in our study.

Upon evaluating the impact of AMS programs on antibiotic use over 6 years, comparing the first period in which only a small proportion of the PICUs had an AMS program with the second period in which 84% had adopted one, in our study we have seen a statistically significant decrease in the antibiotic use ratio and an increase both in the antibiotic-free days ratio and the global antibiotic-free days ratio. Di Pentima et al. [24], lauding the benefits of AMS programs, reported a total decrease of 38% in the use of antibiotics over 3 years in a 180-bed tertiary care pediatric hospital. While this demonstrated the significant impact these programs can have as regards reducing antimicrobial use, this study did not specify the impact on PICUs. Another study by Renk et al. [25] assessed the effect and safety of a once-weekly ward round carried out by pediatric infectious disease physicians, looking at antibiotic consumption in a 14-bed multidisciplinary PICU in Germany, measuring antimicrobial utilization by days of therapy (DoT) per 1000 patient-days (PDs); it reported an 18% reduction of DoT/1000 PD in the period following the implementation of the AMS program. 

As regards the use of broad-spectrum antibiotics, we found that cefotaxime was the most commonly used antibiotic to treat community-acquired infections because of its broad coverage for invasive bacterial disease. Regarding HAIs, the fact that meropenem was more commonly used for outside-PICU-acquired HAIs than for inside-PICU-acquired HAIs (for which the most-used antibiotics were piperacillin–tazobactam and vancomycin) was surprising, as this is mostly used for serious nosocomial infections that are more likely to be contracted in ICUs. This is probably attributable to a greater effort to implement AMS programs in PICUs than in general wards.

The number of antibiotics prescribed empirically was very high (85.6%). Pathogen-targeting therapy was greater for HAIs, most likely due to the poor performance of blood cultures in pediatric patients infected in the community. In fact, when analyzing the rates of appropriate antibiotic use, in 45.7% of the prescriptions it was impossible to detect whether antibiotic therapy was actually appropriate because of negative cultures, with a significantly lower percentage of negative cultures shown for inside-PICU-acquired HAIs.

Comparing the two different periods, we saw a significant increase in terms of appropriate antibiotic use, with a decreasing trend in the use of antibiotics in patients with negative cultures.

An important finding in our study was that up to 20.2% of the antibiotics were discontinued early. In addition, modifications were made to the antibiotic therapy in up to 14.2% of the cases where these were prescribed. The main cause for this adjustment (40.8%) was de-escalation in favor of a narrower spectrum, which was significantly more frequently seen with inside-PICU-acquired HAIs. Ying Guo et al. [26] performed a meta-analysis that reported a de-escalation rate of 39.5% for the antibiotics used in adult patients with severe sepsis and this did not have a detrimental impact on mortality. Furthermore, in a recent publication, Battula et al. [27] reported a de-escalation in 59% of the episodes of sepsis admitted to PICUs and concluded that this strategy appears to be safe.

We also found a statistically significant increase in de-escalation being carried out during the second period, attributable to AMS interventions in physicians’ prescriptions. In line with our observations, Al-Omari et al. [28] stated that just one year after the implementation of AMS programs at four tertiary hospitals, the use of broad-spectrum antibiotics trended downward. García-Rodríguez et al. [29], in a descriptive pre–post interventional study on the use of meropenem, observed that the appropriateness of meropenem prescriptions and the de-escalation from meropenem treatment to narrower-spectrum antibiotics improved progressively over time after an AMS intervention [30,31,32], leading to a significant decrease in meropenem consumption. In our experience, the implementation of comprehensive AMS strategies does modify physicians’ behavior and understanding of the importance of early antibiotic de-escalation and withdrawal as one of the tenets of AMS [25]. 

We acknowledge that our study has some limitations. We report a progressive increase in the number of PICUs that implemented AMS programs; by 2016, 55% of these centers had established such programs. We expect the effects of these programs on antibiotic policies to be less remarkable for those units with a short trajectory.

Another limitation was the fact that we did not assess the impact that maximum PICU occupancy may have on antibiotic prescription, as this phenomenon occurs during the winter season. On the other hand, the sample is very homogeneous because we compared the same months during each year and the healthcare centers included were of the same complexity, with no major changes in staff and patient structure in the PICUs included.

There is a clear association between antibiotic use and antibiotic resistance, but there is much less evidence to support the concept that reducing antibiotic use actually leads to improvements in antibiotic susceptibilities [33]. The clearest and strongest evidence regarding the impact of AMS on antimicrobial resistance is in Clostridium difficile-associated diarrhea [34,35], but there is also increasing evidence suggesting that appropriate antibiotic use can decrease the incidence of resistant Gram-negative bacilli, such as vancomycin-resistant enterococci [36,37].

Antibiotics exert selection pressures on bacterial populations, thus contributing to the development of antibiotic-resistant bacteria. Removing this pressure, however, does not guarantee that the resident microflora will lose the antibiotic-resistance genes. Cook et al. [33] reported the successful implementation of an AMS program, reducing the use of broad-spectrum antibiotics, but saw very little change in the antimicrobial susceptibilities of common Gram-negative pathogens. The assessment of the effect that AMS programs have with respect to changing bacterial susceptibilities remains a challenge for us to tackle in the future. 

AMS programs may be considered as pivotal contemporary healthcare policies which have some of the greatest impacts on both patient outcomes and healthcare costs. They should thus be a top priority for all healthcare providers, otherwise there may be serious long-term impacts whose ramifications we cannot currently comprehend. In today’s clinical environment, especially in pediatric areas, antimicrobials are being unnecessarily prescribed to treat mainly viral infections, which implies not only an unnecessary cost but a risk to patients as well as to public health.

We have already emphasized the importance of antibiotic stewardship programs in PICUs. Our results show that AMS programs are related to prescribing appropriate antibiotics and also to reducing the use of broad-spectrum antibiotics, which could be the key to slowing the development of antimicrobial resistance. Moreover, after the implementation of AMS programs, we found a statistically significant increase in the number of antibiotics that were adjusted to target a narrower spectrum, which is a practice that has consequences for public health.

## 5. Conclusions

Our study highlights the importance of implementing AMS programs, which have been shown to be effective in reducing antimicrobial utilization, altering prescribing behavior and encouraging the appropriate use of antibiotics. This will help our healthcare systems better tackle both current and future challenges, such as controlling antimicrobial resistance. Further studies are required on clinical outcomes, hospital costs, indirect expenses such as antibiotic side effects, earlier transitions to oral therapy, length of hospital stay, and readmissions. In addition, future long-term studies are needed that focus mainly on mortality and antimicrobial resistance in the pediatric population.

## Figures and Tables

**Figure 1 children-09-00902-f001:**
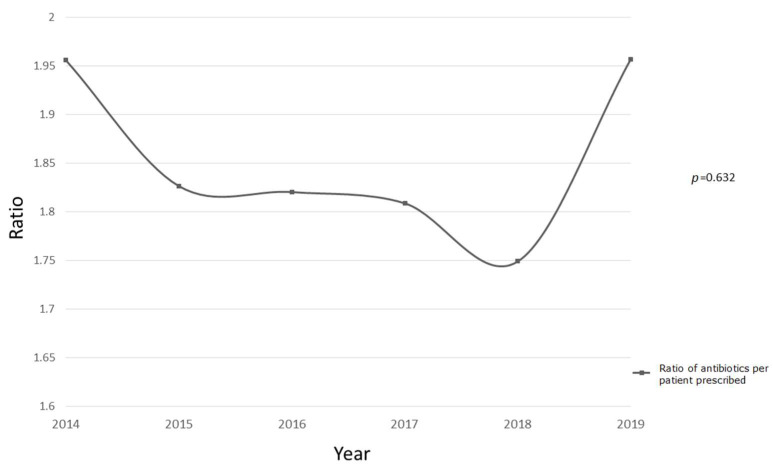
Evolution of the ratio for the number of antibiotics per patient prescribed antibiotics. Comparison of proportions between 2014–2016 vs. 2017–2019 expressed as *p*-values.

**Figure 2 children-09-00902-f002:**
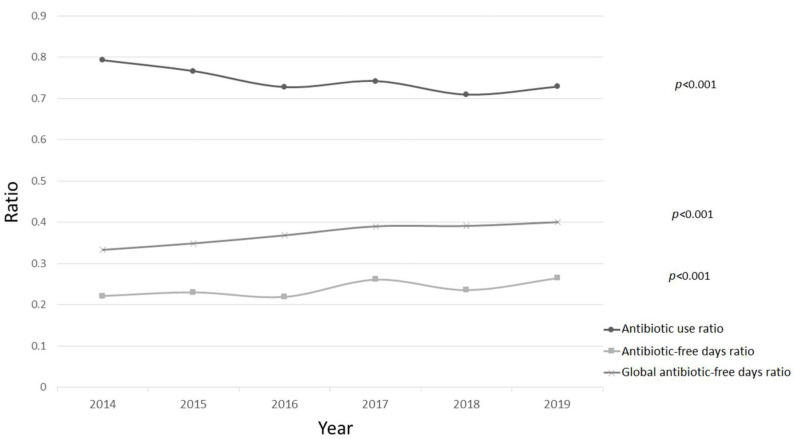
Representation of the evolution of different ratios regarding the use of antibiotics. Comparison of proportions between 2014–2016 vs. 2017–2019 expressed as *p*-values.

**Figure 3 children-09-00902-f003:**
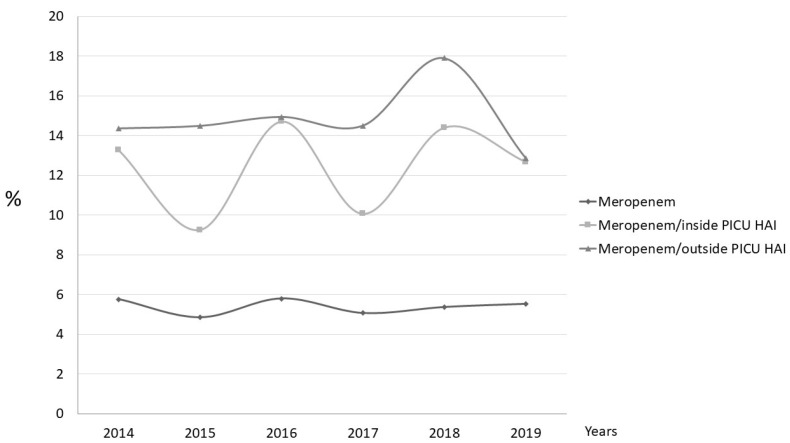
Evolution over time of the global use of meropenem (meropenem with respect to the global antibiotic indication) and the use of meropenem for suspected healthcare-associated infection (prescription of meropenem with respect to the indication of antibiotics for suspected healthcare-associated infection).

**Figure 4 children-09-00902-f004:**
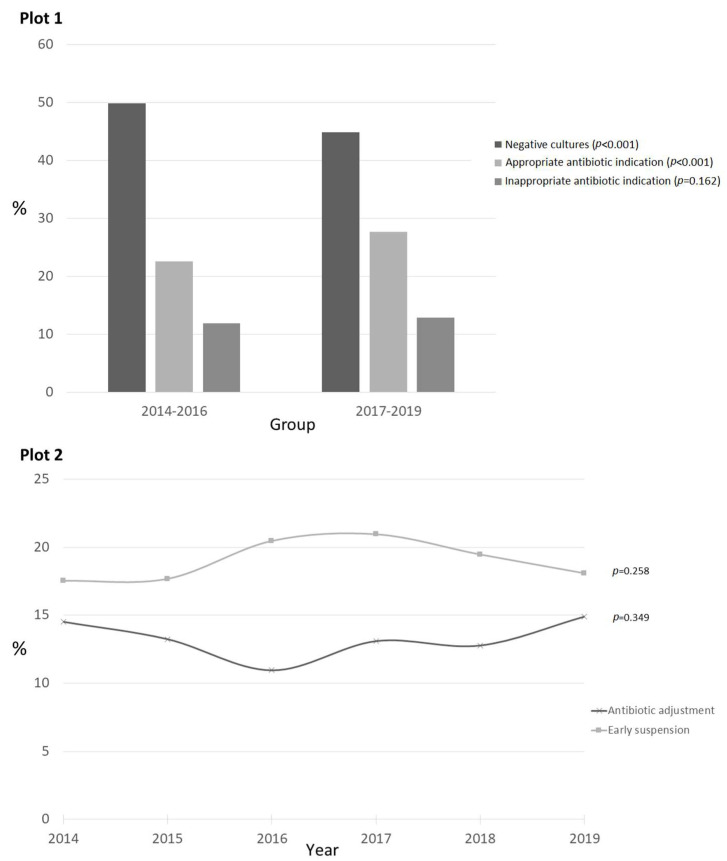
Regarding empirical antibiotic prescriptions, Plot 1 compares the accuracy of the antibiotic indications between the two time periods (2014–2016 vs. 2017–2019). It includes the proportion of appropriate antibiotic indications, inappropriate indications, and cases with negative cultures. Plot 2 represents the evolution over time of the need for antibiotics to be adjusted and suspended early in empirical indications.

**Figure 5 children-09-00902-f005:**
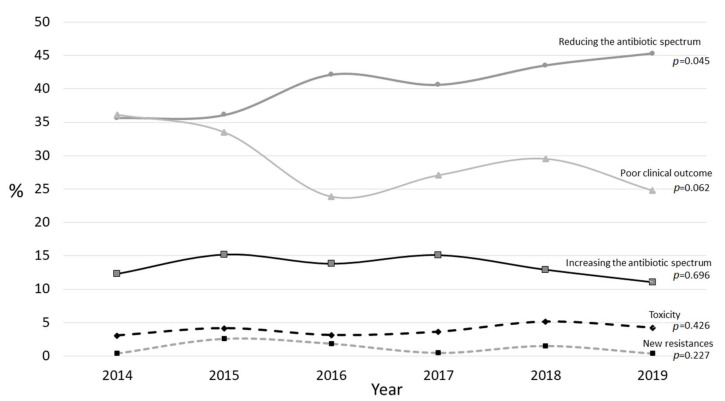
Representation of the evolution of the different reasons for switching antibiotics in those patients who required an adjustment of their antibiotic therapy.

**Table 1 children-09-00902-t001:** Definitions of healthcare-associated infections according to ENVIN-Ped and based on CDC definitions [11,12].

Type of Infection	Definition
Central line-associated bloodstream infection (CLABSI)	Primary blood stream infection (no other apparent source of infection) and positive blood cultures, all involving the same microorganism, fulfilling one of the following situations: Quantitative central venous catheter (CVC) culture ≥10^3^ CFU/mL; Quantitative blood culture ratio CVC blood sample/peripheral blood sample >5; Differential delay in positivity of blood cultures: CVC blood sample culture positive two hours or more before peripheral blood culture; Positive culture with the same microorganism found in pus at insertion site.
Ventilator-associated pneumonia (VAP)	A. Clinical diagnosis: Presence of a new and persistent pulmonary infiltrate on one chest X-ray or CT scan in a previously healthy patient; OR Two or more images suggestive of pneumonia in patients with underlying heart or lung disease; AND at least one of the following: Fever ≥ 38 °C with no other discernable cause; Leukopenia (<4000 WBC/mm^3^) or leukocytosis (≥12,000 WBC/mm^3^); AND at least one of the following (or at least two, if clinical pneumonia only): Increased respiratory secretions, change in previous characteristics of sputum, or sputum with purulent appearance; New onset of cough, dyspnea, and/or tachypnea; Abnormal lung sounds, such as crackles, bronchial breath, wheezing; Increased oxygen requirements or ventilatory demand; AND, depending on the diagnostic method used: B. Bacteriological diagnosis: (PN1). Positive quantitative culture from a minimally contaminated specimen: Bronchoalveolar lavage (BAL) with a threshold of ≥1 × 10^4^ colony-forming units (CFU)/mL or ≥5% of BAL-obtained cells containing intracellular bacteria upon direct microscope exam; Protected specimen brush or protected distal aspirate, with a threshold of ≥1 × 10^3^ CFU/mL; (PN2). Positive quantitative culture from a possibly contaminated specimen: Quantitative culture from an endotracheal aspirate with a threshold of ≥1 × 10^6^ CFU/mL; (PN3). Alternative microbiological methods: Positive blood culture not related to another source of infection; Positive growth in pleural fluid culture; Pleural or pulmonary abscess, with positive needle aspiration; Histological evidence of pneumonia; Positive detection of viral antigen or antibodies in respiratory secretions; Seroconversion; or Detection of viral antigen in urine; (PN4). Positive sputum culture or non-quantitative specimen culture; (PN5). No positive microbiology.
Catheter-associated urinary tract infections (CAUTI)	Defined in a patient who has at least one of the following symptoms, with no other recognized cause: Fever >38 °C, increased urgency and/or frequency, dysuria, or suprapubic tenderness; Pyuria in urine specimen, with ≥10 WBC/mL or ≥3 WBC/high-power field of unspun urine; AND Positive urine culture with a threshold of ≥1 × 10^5^ CFU/mL with no more than two species of microorganisms in a patient that is not receiving antibiotic treatment; OR A threshold of <1 × 10^5^ CFU/mL of one single microorganism in patients receiving antibiotic treatment.

**Table 2 children-09-00902-t002:** Description of the sample.

Variables	Global	2014	2015	2016	2017	2018	2019	2014–2016	2017–2019	*p*-Value
Hospitals (*n*)	33	27	27	25	24	29	26	79	79	-
Total admissions (*n*)	11,260	1724	1748	1877	1983	2176	1752	5349	5911	-
Stays (days)	69,512	11,743	11,635	10,972	11,556	12,880	10,726	34,350	35,162	-
LOS in days, median (*IQR*)	3 (6–2)	4 (3–7)	4 (3–6)	3 (2–6)	3 (2–6)	3 (2–5)	3 (2–6)	4 (2–7)	3 (2–6)	<0.001
Age in months, median (*IQR*)	43 (10–115)	42.3 (9.6–104.9)	40 (8–111)	47.7 (11–118)	42 (9–115)	42 (10–121)	46 (11–120.3)	43 (10–111)	43 (10–119)	0.096
Gender (male), *n* (%)	6368 (56.6%)	971 (56.3%)	1019 (58.3%)	1057 (56.3%)	1121 (56.5%)	1242 (57.1%)	958 (54.7%)	3047 (57%)	3321 (56.2%)	0.404
PRISM score, median (*IQR*)	2 (0–5)	3 (0–7)	3 (0–7)	2 (0–5)	2 (0–5)	2 (0–6)	2 (0–5)	3 (0–6)	2 (0–5)	<0.001
Comorbidity, *n* (*%*)	2317 (20.6%)	396 (23%)	380 (21.7%)	377 (20.1%)	419 (22.1%)	388 (17.8%)	357 (20.4%)	1153 (21.6%)	1164 (19.7%)	0.015
New AMS programs, *n* (*%*)	26 (84%)	9 (29%)	4 (12.9%)	4 (12.9%)	1 (3.2%)	6 (19.4%)	2 (6.4%)	17 (54.8%)	9 (29%)	-
HAI/1000 patient-days (*‰*)	6.3	7.3	5.9	6.7	5.1	5.4	7.5	6.6	6	-
Deaths, *n* (*%*)	213 (1.9%)	44 (2.6%)	35 (2%)	45 (1.8%)	35 (2%)	30 (1.4%)	24 (1.4%)	124 (2.3%)	89 (1.5%)	0.002

**Table 3 children-09-00902-t003:** Comparison between the two periods regarding the different reasons for antibiotic indications.

Type of Antibiotic Indication, *n* (%)	2014–2016 *n* = 7597	2017–2019 *n* = 7851	*p*-Value
Suspicion of community-acquired infection	2328 (30.6%)	2362 (30.1%)	0.451
Suspicion of outside-PICU-acquired HAIs	1075 (14.2%)	1047 (13.3%	<0.001
Suspicion of inside-PICU-acquired HAIs	1059 (13.9%)	1142 (14.5%)	0.281
Surgical prophylaxis	2447 (32.2%)	2577 (32.8%)	0.416
Non-surgical prophylaxis	613 (8.1%)	686 (8.7%)	0.134
Unknown reason for prescription	75 (0.99%)	37 (0.47%)	<0.001

**Table 4 children-09-00902-t004:** Use of antibiotics by suspected infection indication.

Ratios	2014	2015	2016	2017	2018	2019	2014–2016	2017–2019	*p*-Value
No. of antibiotics per patient prescribed antibiotics									
All indications	1.96	1.83	1.82	1.81	1.75	1.96	1.87	1.84	0.632
Community-acquired infections	1.95	1.81	1.90	1.82	1.83	1.87	1.89	1.84	0.425
Outside-PICU-acquired HAIs	2.35	2.52	2.59	2.55	2.28	2.32	2.49	2.38	0.111
Inside-PICU-acquired HAIs	2.51	2.84	2.59	2.49	2.54	3.64	2.65	2.89	<0.001
Antibiotic use ratio									
All indications	0.79	0.77	0.73	0.74	0.71	0.73	0.76	0.73	<0.001
Community-acquired infections	0.24	0.24	0.22	0.20	0.21	0.24	0.23	0.22	0.137
Outside-PICU-acquired HAIs	0.09	0.08	0.08	0.08	0.06	0.09	0.08	0.08	1.000
Inside-PICU-acquired HAIs	0.09	0.07	0.06	0.07	0.07	0.07	0.07	0.07	1.000
Antibiotic-free days ratio									
All indications	0.22	0.23	0.22	0.26	0.24	0.26	0.22	0.25	<0.001
Community-acquired infections	0.15	0.21	0.28	0.24	0.28	0.29	0.21	0.27	<0.001
Outside-PICU-acquired HAIs	0.33	0.28	0.25	0.26	0.29	0.36	0.29	0.30	0.173
Inside-PICU-acquired HAIs	0.47	0.46	0.45	0.53	0.47	0.49	0.46	0.50	<0.001
Global antibiotic-free days ratio									
All indications	0.33	0.35	0.37	0.39	0.39	0.40	0.35	0.39	<0.001
Community-acquired infections	0.70	0.78	0.80	0.83	0.78	0.79	0.76	0.80	<0.001
Outside-PICU-acquired HAIs	0.88	0.89	0.90	0.89	0.93	0.89	0.89	0.90	0.043
Inside-PICU-acquired HAIs	0.84	0.85	0.86	0.87	0.86	0.49	0.85	0.74	<0.001

**Table 5 children-09-00902-t005:** Summary of the most commonly used antibiotics for the different indications.

	Total *n*, %		2014–2016 *n*, %		2017–2019 *n*, %		*p*-Value
Surgical prophylaxis	5024		2447		2577		
Cefazolin	2457	48.9	1119	45.7	1338	51.9	<0.001
Amoxicillin–clavulanate	788	15.7	381	15.6	407	15.8	0.828
Community-acquired infections	4690		2328		2362		
Cefotaxime	1097	23.4	548	23.5	549	23.2	0.810
Amoxicillin–clavulanate	510	10.9	251	10.8	259	11.0	0.840
Vancomycin	368	7.8	187	8.0	181	7.7	0.638
Outside-PICU-acquired HAIs	2122		1075		1047		
Meropenem	313	14.8	157	14.6	156	14.9	0.739
Vancomycin	230	10.8	86	8.0	144	13.8	<0.001
Piperacillin–tazobactam	167	7.9	109	10.1	58	5.5	<0.001
Amikacin	139	6.6	68	6.3	71	6.8	0.928
Teicoplanin	90	4.2	29	2.7	61	5.8	<0.001
Inside-PICU-acquired HAIs							
Vancomycin	346	15.7	164	15.5	182	15.9	0.772
Piperacillin–tazobactam	340	15.4	162	15.3	178	15.6	0.851
Meropenem	273	12.4	131	12.4	142	12.4	0.964
Teicoplanin	166	7.5	64	6.0	102	8.9	0.010
Amikacin	135	6.1	71	6.7	64	5.6	0.282

Categorical variables expressed as frequencies (percentages) and compared using the χ^2^ test. PICU: pediatric intensive care unit, HAI: healthcare-associated infection.

## Data Availability

Not applicable.

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
