# Peer review of "Antimicrobial Stewardship Improvement in Pediatric Intensive Care Units in Spain—What Have We Learned?"

_children, 2022, doi:10.3390/children9060902_

Round 1
Reviewer 1 Report
The authors investigated antimicrobial prescriptions, antibiotic indications, and antibiotic stewardship in PICU in Spain. The flow of the results is too difficult to read, making the message of this manuscript vague.
Major comments
- English editing is strongly recommended. For example,
Line 25; increased
Line 269; broaden
- Original definitions of this study were used in the Abstract without explanations (e.g., free-days antibiotic ratio and global free-days antibiotic ratio). This could be bothersome to readers. It should be modified.
- I could not understand the descriptions in Lines 98-106. Please rephrase. The paragraph in Lines 295-302 seems appropriate here, not in Discussion. I suggest to add the percentage of AMS programs in Table 2.
- Can the authors rephrase free-days antibiotic ration and global free-day antibiotic ratio (Lines 133-137)? Difficult to understand.
- Patient-days are frequently used in this kind of the study. Are data presented in this manuscript relevant and possible to compare the previous similar studies?
- The incidence of HAI in this study are described in Lines 159-162. Are they more or less compared to the other studies in PICU?
Minor comments
- Lines 46-53; they seemed author guide, not the manuscript. They should be removed.
- Lines 242; what’s the definition of “adequate”?
Author Response
Dear reviewer,
Thank you for your contributions. I am sure that they will improve the content of the article.
Best regards.

Reviewer 2 Report
In this study, antibiotic stewardship was assessed in the pediatric intensive care units in Spain. This is an important topic, and interesting analyses were performed. However, some features should be answered before potential publication:
- Line 46-54 – this part of the Introduction is just explanation how it should be written and must be removed
- Furthermore, in the Introduction section, better emphasis should be put into gap in the literature and novelty in the field this study is bringing, as well as main hypotheses authors chose to investigate
- Line 92-93 – how and why was this exact date range for each year selected? Rationale needed in detail
- In Discussion section, more emphasis should be on potential clinical or management implications that can be derived from the results of this study
Author Response

(The authors gave the same response as above.)

Round 2
Reviewer 1 Report
Point 3: I could not understand the descriptions in Lines 98-106. Please rephrase. The paragraph in Lines 295-302 seems appropriate here, not in Discussion. I suggest to add the percentage of AMS programs in Table 2.
- We have added the percentage of new hospitals that implemented AMS programs each year in Table 2.
→”new” should be included in Table 2 for clarification.
Point 6:
The incidence of HAI in this study are described in Lines 159-162. Are they more or less compared to the other studies in PICU?
Response 6:
If we compare the incidence with previous studies in PICU of low and middle income countries, we appreciate a wide difference in the HAI rates (higher than ours). However we have found these two recent studies in PICU showing rates comparable to ours (Greece and Jordan). Should we add them in the references?
Briassoulis P, Briassoulis G, Christakou E, Machaira M, Kassimis A, Barbaressou C, Nikolaou F, Sdougka M, Gikas A, Ilia S. Active Surveillance of Healthcare-associated Infections in Pediatric Intensive Care Units: Multicenter ECDC HAI-net ICU Protocol (v2.2) Implementation, Antimicrobial Resistance and Challenges. Pediatr Infect Dis J. 2021 Mar 1;40(3):231-237. doi: 10.1097/INF.0000000000002960.
Elnasser Z, Obeidat H, Amarin Z. Device-related infections in a pediatric intensive care unit: the Jordan University of Science and Technology experience. Medicine 2021;100:43(e27651).
→Yes, I recommend to add the new references, because they can make understand the place of this study in this field.
Point 8:
Lines 242; what’s the definition of “adequate”?
Response 7:
Sorry for the misunderstanding. We changed from “adequate” to “appropriate”, which means that the antibiotic provides the right treatment for the infection (Lines 299-313 and Figure 3).
→Definition is necessary for “appropriateness”. This is an example from previous similar study.
“Empiric therapy was considered effective when the bacterium was susceptible in vitro to the antibiotics.”
Author Response
Dear reviewer,
Thank you for your contributions.
Best regards.

Reviewer 2 Report
The authors have addressed the comments and therefore improved the quality of the manuscript. I have no further questions.
Author Response

(The authors gave the same response as above.)
